# Patient Education in Spa Resorts: Experience from a French National Program for Patients with Chronic Venous Insufficiency

**DOI:** 10.3390/ijerph19031176

**Published:** 2022-01-21

**Authors:** Patrick H. Carpentier, Bernadette Satger, Brigitte Sandrin

**Affiliations:** 1Centre de Recherche Universitaire de La Léchère, Université Grenoble-Alpes, 73260 La Léchère, France; besatger@free.fr; 2Association Française pour le Développement de l’Education Thérapeutique 18, Passage de la Bonne Graine, 75011 Paris, France; brisanber@gmail.com

**Keywords:** venous insufficiency, therapeutic education, patient education, balneotherapy, spa resort, health resort

## Abstract

The prognosis of chronic venous insufficiency (CVI) is greatly depending upon the ability of the patients to optimize their health-related behaviors (mainly compliance to compression stockings, physical activity and diet). As this is usually challenging for the patients, we developed a therapeutic education program (TEP) aiming at helping them to achieve these optimizations. We report here the preliminary results obtained with this program. This structured TEP was developed by a working group of 15 health professionals to be used during the regular French spa 3-week treatment course for CVI patients. The program included three interactive workshops aiming at improving the patients’ knowledge, skills and motivation, two educational consultations allowing to set and follow-up their personal action plans and a built-in evaluation. It was implemented in spa resorts specialized in CVI. Among the first 150 patients (116 women and 34 men, mean age 69 years old (SD 8 years), 49% had post-thrombotic disease. Compliance to the education workshops was 98%. After a 3-month follow-up, 83% of the patients fully achieved at least one behavioral objective. Quality of life, as assessed by the CIVIQ 2 auto-questionnaire, improved at 3 months (*p* = 0.0024) and 9 months (*p* = 0.018). These results demonstrate the feasibility of a TEP for CVI patients and its ability to improve their health-related behaviors, opening the way for the development of similar programs for other chronic diseases in the setting of spa resorts.

## 1. Introduction

Chronic Venous Insufficiency (clinical classes C4 to C6 of the CEAP international classification of Chronic Venous Disorders (CVD)) [1] is a disabling condition substantially altering the patients’ quality of life. Its prevalence is high (4–9% [2,3]) in the general population, with high costs for the health services in industrial countries [2]. In particular, patients with C4 (pigmentation, dermatitis or lipodermatosclerosis) or C5 (history of leg ulcer) have a high risk of leg ulcer incidence (C6) [4]. It has been shown that lifestyle and health related behaviors such as overweight, lack of physical exercise, and compliance to compression therapy represent important risk factors for the worsening of the condition and incidence of leg ulcers [4,5].

Although the correction of the abnormal venous hemodynamics is important and feasible in patients with CVI and incompetence of the superficial venous system and varicose veins, this may be insufficient in patients with already altered skin microcirculation, and for those with a predominance of deep vein system incompetence. In patients with those conditions, rehabilitation and the improvement of the patients’ health-related behavior are valuable therapeutic targets as stated in international guidelines [6,7,8].

It is striking, however, that very few attempts have been carried out in order to empower the patients with severe CVD in the management of their disease [9,10] as it was developed for patients with other chronic diseases such as asthma or diabetes mellitus. Twenty years ago, we started to try alleviating this situation setting-up an empirical program teaching the patients about their disease in a French spa-resort dedicated to the care of patients with CVI, which was already well received [11]. When performing a randomized trial evaluating the efficacy of the balneotherapeutic treatment in these patients, it was combined with this former educational approach, and we were able to show a statistically and clinically significant effect of this treatment [12], which was later confirmed in a larger multicenter randomized trial involving 425 patients [13]. One striking feature in our findings was the long duration—at least twelve months—of the patients’ improvement. We then hypothesized that this sustained duration was related to changes in the health-related behaviors of the subjects, and therefore decided to invest in a more structured therapeutic education program fulfilling the quality criteria of the WHO and from the French National Health Agency (Haute Autorité de Santé) [14]. We took the opportunity of the large number of CVI patients (close to 100,000 each year) benefiting in France from a balneotherapeutic rehabilitation treatment performed in spa resorts dedicated to this indication to set-up this new program in a favorable environment. The aim of this paper is to report our preliminary experience with this program and the effects observed on patient health-related behavior and quality of life, as shown in the first 150 patients enrolled in the program. To our knowledge, this is the first report of a structured Therapeutic Education Program (TEP) in the field of CVI.

## 2. Materials and Methods

### 2.1. Patients

The educational program was proposed to the patients who were receiving the Spa treatment in one of the three participating spa resorts of Argelès-Gazost, La Léchère and Barbotan. They had to volunteer for the program and to have a Chronic Venous Insufficiency with skin changes corresponding to the C4 class of the CEAP classification (pigmentation, dermatitis or lipodermatosclerosis) or a history of venous leg ulcer (C5) but no active ulcer.

### 2.2. Patient Therapeutic Education Program 

The education program was built-up by a national pluri-professional working group of 15 health professionals from the 12 French spa resorts accredited for CVD patients with the help of a group of patients who gave their feed-back regarding the proposed educational objectives and tools. 

In each spa resort, it was applied by a local team of health professionals, who had benefited from a three-day training in patient education specifically dedicated to this program. The program included, for each patient: three interactive workshops for small groups of patients (6 to 8 patients), a face-to-face individual education consultation with educational diagnosis and setting-up with the patient of a personalized action plan including one to three health-related behavioral objectives and discussion about the way to achieve them, and a second educational consultation for personalized follow-up through phone interview. 

The three workshops used different kinds of group facilitation techniques under the supervision of a health care professional trained in patient education. They were aiming at increasing the knowledge of the patient about his disease and its treatments (workshop 1), at improving his understanding of the influence of his lifestyle on his/her venous system (workshop 2) and at developing skills with the handling of the compression stockings (workshop 3); a special attention was paid to increasing the patients’ motivation for an active participation in their treatment. 

It was taken advantage of the associated balneology cares, which provided the patient with an opportunity to experience repetitively the physical effects of pressure, movement related massages, temperature changes etc., helping him understanding and adhering to the importance of compression therapy, physical activity, their combination and so forth.

### 2.3. Balneotherapeutic Treatment 

The spa treatment regimen consisted of four balneology sessions per day, 6 days a week during 3 weeks. These sessions were customized for each patient by a specialized spa physician according to the patient’s needs and capabilities, on arrival at the spa resort as required by French health authority rules. Typically, these sessions most often included [13]:-a 15-min walking session in a specially designed pool with walking corridors in semi-deep (80 cm) cool (28 degrees Celsius) water (training of muscle pump function);-a 20-min whirlpool bath session with automatic air and water massage cycles (aiming at mobilization of the superficial skin volume blood flow);-a 10-min bath session with customized underwater strong massaging jets (softening of the sclerotic subcutaneous tissues);-and a 10-min massage session of the leg and ankle skin areas by a registered physiotherapist under a light spray shower (softening of the sclerotic subcutaneous tissues) or a 15-min joint mobilization session in a deep (150 cm) warm (34 degrees Celsius) pool under the supervision of a physiotherapist (improvement of joint mobility for better ambulation and calf muscle venous pump functioning).

As mentioned above, balneology cares were used as adjunctive educational tools for a better understanding and motivation to apply the most important educational messages. And the educational workshops also motivated the patients for a more active participation in their care sessions in a most synergistic way.

### 2.4. Outcome Assessment

The therapeutic education program had a built-in evaluation required by the French Haute Autorité de Santé [14]. We observed retrospectively and analyzed the evaluation data of the 150 first consecutive patients in order:-to confirm the feasibility and acceptability of the program (participation rate to the workshops and satisfaction questionnaire at day 18);-to evaluate the full or partial achievement at day 90 of the one to three health-related behavioral objectives decided by the patient during the education course;-to measure the parallel variation in CVI related quality of life, using the CIVIQ 2 auto-questionnaire specifically developed for patients with CVI with world-wide recognition [15] (day 0, day 90 and day 270).

### 2.5. Statistical Analysis

Descriptive statistics are given as mean (SEM). There was no control group, but before/after comparisons were performed in order to show the magnitude of the changes observed. They used paired Wilcoxon tests. A *p* < 0.05 alpha risk threshold level was considered significant.

## 3. Results

### 3.1. Characteristics of the Study Patients

150 patients with advanced venous insufficiency (C4 to C5; 116 women and 34 men, aged 69 years old (SD 8 years) were enrolled in the program in the three French spa resorts specialized in chronic venous disorders (Argelès-Gazost, Barbotan and La Léchère). Their main characteristics are given in Table 1. 

### 3.2. Compliance to the Therapeutic Education Program

They followed the patient education program and spa treatment with a high compliance (98% mean attendance to the 3 educational workshops). During the individual education consultations (mean duration 56 ± 4 min), they decided to achieve one (96%), two (85%) or even three (43%) target objectives regarding health-related behavioral changes in their everyday life.

### 3.3. Satisfaction Level

At the end of the education program (day 18), the satisfaction level of the participants was very high as shown by their agreement with the proposed sentences listed in Table 2. Most of them thought they had acquired useful knowledge (91%), but even more were feeling they had increased their ability to cope with their disease (97%), and that they would become more active in its management (94%).

### 3.4. Main Results: Improvement in Health-Related Behaviors

After a 3-month follow-up, the evaluation by phone interview of the 143 patients who decided for at least one behavioral objective (one drop-out and one answer refusal) showed that 61% of the objectives were fully achieved, 21% partially obtained and 18% unreached (Table 3). In total, 83% of the patients fully achieved at least one behavioral objective, and 51% fully achieved two or three; only 4% did not accomplish any measurable change in behavior.

### 3.5. Improvement in Health-Related Quality of Life

In the meantime, as shown in Figure 1, the CIVIQ 2 score improved in 62% of the patients at 3 months (paired Wilcoxon: *p* = 0.0024), and 59% at 9 months (paired Wilcoxon *p* = 0.018).

## 4. Discussion

The most striking result of this preliminary evaluation program is the very high level of satisfaction and motivation shown by the patient at the end of the program course (Table 2); this enthusiasm made almost all of them able to decide at least one and most often two or three health-related behavioral changes in their everyday life. This is certainly related to the positive personal interactions during the workshops and the personalization of the program. But we believe that the balneotherapeutic cares, especially those involving an active participation of the patients such as the walking pool, reinforces the impact of the educational messages. And the 3-week stay in a spa positive resort environment also helps the patient acquiring a positive mind set.

The achievement of health-related behavioral changes decided during the therapeutic program is not perfect, but indeed, a majority of patients achieved at least two objectives and a very small minority (4%) did not make any improvement. We feel this is remarkable since improving his level of physical activity, controlling his diet, and even more improving his compliance to compression stockings are not easy changes to make. And even more so to stick to these changes for as long as a 3-month follow-up, the usual term in the field of therapeutic education for the demonstration of long-lasting impact.

The observed improvement of disease-related quality of life, as measured by the CIVIQ 2 score, is probably resulting from the education program but also from the balneotherapeutic treatment, which has already shown such an effect in randomized controlled trials [9,10]. An evaluation of the impact on the clinical status itself (skin changes), and not only on quality of life would be important too, but the longer observation time course needed was beyond the possibilities of this work.

However, the main limitation of this study is that it is only observational, with no control group, and although very encouraging, cannot be interpreted as definitively demonstrating a therapeutic effect of the education program by itself.

It is important to note that our therapeutic education program was designed to complement a balneotherapeutic stay dedicated to the rehabilitation of patients with chronic venous insufficiency. The specific health-resort atmosphere of spa resorts is certainly helpful for increasing the patient ability to adhere to such a program, and the large number of patients with similar conditions present at the same time in such a resort is facilitating the organization of group educational activities at reasonable cost.

This kind of program could certainly be used in other settings such as rehabilitation centers, health education centers or other outpatient facilities. However, as in a country such as France, approximately 1% of the whole population get a spa treatment course each year for one or another chronic disease, the opportunity to develop similar programs in these particular settings for patients with rheumatic, pulmonary, dermatological or metabolic chronic diseases should certainly be considered.

## 5. Conclusions

This study illustrates the high positive impact of a therapeutic education program performed in the setting of a spa treatment on the health-related behaviors of the patients with C4-C5 chronic venous insufficiency. These results allowed the accreditation of the program by the French National Health System, on a national basis, for patients with advanced chronic venous insufficiency. These encouraging results need a confirmation and a quantification by a randomized controlled trial.

## Figures and Tables

**Figure 1 ijerph-19-01176-f001:**
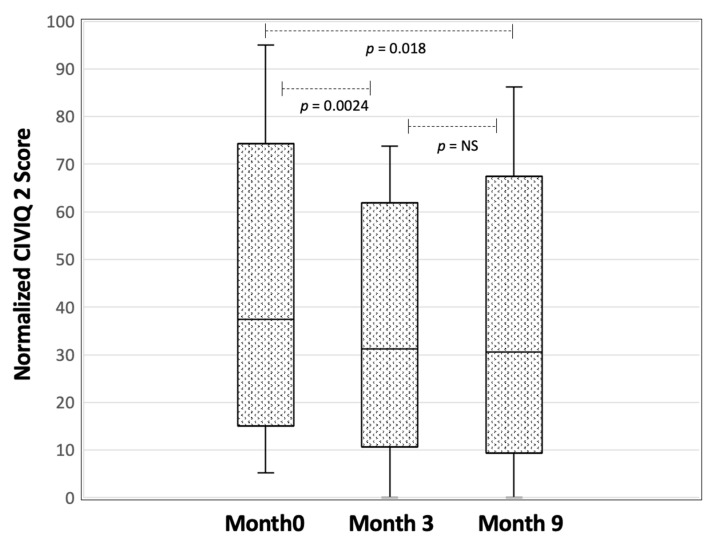
Evolution of normalized CIVIQ 2 score at months 3 and 9 after education course. A high score means a reduced quality of life.

**Table 1 ijerph-19-01176-t001:** Main characteristics of the patients enrolled in the three study sites.

	Argelès (*n* = 32)	Barbotan (*n* = 85)	La Léchère (*n* = 33)
Sex (% females)	61%	80%	85%
Age (years; mean ± SD)	68 ± 11	70 ± 7	69 ± 8
C4 (most affected leg)	95%	74%	81%
C5 (most affected leg)	5%	24%	19%
Post-Thrombotic Syndrome	81%	38%	55%
Anticoagulant therapy	44%	10%	24%

**Table 2 ijerph-19-01176-t002:** Patients’ perception regarding the education program at day 18 (149 answers–1 dropout).

	How Well Do You Agree with the Following Statements?			
	Not at All	Rather No	Rather Yes	Quite So
“I am happy with my participation to the program”	0%	0%	25%	75%
“I learned useful knowledge about my venous disease”	3%	6%	43%	48%
“I feel I have increased my ability to live well with my disease”	1%	2%	38%	59%
“I think I will be more active in the management of my disease”	1%	5%	37%	57%

**Table 3 ijerph-19-01176-t003:** Achievement of health-related behavior objectives at day 90 (143 answers; 2 dropouts and 5 patients preferring not to set goals).

Behavior Changes	Fully Achieved	Partly Achieved	Unreached
Objective 1 (*n* = 143)	59%	25%	15%
Objective 2 (*n* = 125)	66%	15%	19%
Objective 3 (*n* = 64)	56%	21%	23%

## Data Availability

The data presented in this study are available on request from the corresponding author.

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
