# Peer review of "Patient Education in Spa Resorts: Experience from a French National Program for Patients with Chronic Venous Insufficiency"

_ijerph, 2022, doi:10.3390/ijerph19031176_

Round 1
Reviewer 1 Report
Thank you for this interesting paper that review using a therapeutic education program to improve chronic venous insufficiency. My questions are:
- What are the cost of TEP? Is it readily available for most patients?
- Is the improvement in quality of life related to education or the specific therapies that was provided in the spa
- How does this apply to the general population who do not have access to spa therapy
Reviewer 2 Report
Thank you for submitting a manuscript entitled “Patient Education in Spa Resorts. Experience from a French national program for patients with chronic venous Insufficiency”. My comments are as follow.
Abstract:
- The authors should follow the format of the Journal
- What are C4-5 patients?
- Line 21, aged 69+8 meant mean age 69 and SD8?
- Conclusion should include the message if the objectives were achieved and any suggestions to current healthcare settings or to future research.
Introduction
- Need to follow the format of the journal mentioned author’s guideline.
- The introduction part needs to have more elaboration. The argument about the need of the educational program is not enough. The research gap is not clear.
- A section of the literature review regarding the interventions for CVD or CVI patients in the current practice is necessary. What happened to these interventions and why did not conduct this research?
- The title mentioned Chronic venous insufficiency, but this term was not mentioned in the whole introduction. CVI is under the term CVD. What happened to patients with CVI? The authors need to elaborate more
- What did C stand for? Eg 28C, 34C? What are they?
Materials and methods
- The authors should mention the study design….following the author’s guideline
- What is CEAP? It needs more explanation.
- Regarding educational programme, how many sessions? How often was each session given? How to manage the absentee(s) in any session of the programme?
- In evaluation, the authors should explain how to evaluate each workshop according to the needs of psychological distress with CVI
- Statistics were not enough if several study periods were included but only using paired Wilcoxon tests. Did the authors use other analytic methods to understand the differences of the outcome measures between/among study periods? If so, please report.
Results
- The authors should follow the journal format.
- The results were not presented thorough using the statistical methods mentioned (ie paired Wilcoxon tests.
Discussion
- Line 168, the results were presented as “very high level(s) of satisfaction…” but actually not unclear according to the results.
- Line 179, the authors mentioned that the effect of the educational programme was sustained with a long-lasting impact after a 3 month followup. Was there any immediate post educational data to compare the effects of programme with that at the 3-month? The effects were not significant at month 9, could you claim that was long-lasting effect?
- The results were presented too brief. This longitude study should have more fruitful results to let audience understand more about the effect of the programme on outcome measures.
- The citations are not enough to support your result interpretation. It can be due to inadequate presentation.
Overall
The content in each part is inadequate to be presented. Authors need to review each section, follow the Journal format, elaborate more information/explanation.
Further data analysis and data presentation should be done appropriately to achieve the study objectives.
English proficiency should be checked.
Reviewer 3 Report
The MS: ”Patient Education in Spa Resorts. Experience from a French national program for patients with chronic venous Insufficiency” is interesting and helpful.
The Authors may consider some suggestions:
In the discussion, the limitations of the study should be included as a separate paragraph (e.g.,
“This study is only observational, with no control group, and cannot be interpreted as definitively demonstrating a therapeutic effect of the therapeutic education program by itself.”).
In the “future research directions” in this area, with regard to some helpful tools to evaluate the patient’s progress over time, the following information may be added:
The CEAP classification allows putting patients into particular categories of their CVD. However, to evaluate the patient’s real progress over time, some instruments that can illustrate changes, as the patient is being treated are needed. For instance, the VCSS (Venous Clinical Severity Score), which has 9 components of CVD - from simple leg pain and edema up to ulceration, can be used when patients are initially seen, and then the VCSS can be repeated, to assess how he or she is doing overtime. Unfortunately, the VCSS (that is based on the physician’s determined scoring) does not consider what is important from the patient’s perspective.
It should be underscored that in a patient-reported outcome tool, such as the VVSymQ™ the questions are generated by patients who suffer from CVD. The VVSymQ™ reports on the 5 most common symptoms that patients with CVD experience: heaviness, aching, swelling, throbbing, and itching. Importantly, this is a scoring system presented from the patient’s point of view, which can be used to evaluate how a given therapy has helped. In contrast to the CEAP classification, both the VCSS score and the VVsymQ™ can change more dynamically, in response to the applied treatments. This may be particularly useful for patient education.
Round 2
Reviewer 2 Report
Thank you for submitting your manuscript entitled ‘Patient Education in Spa Resorts. Experience from a French national program for patients with chronic venous Insufficiency’. My comments can be found below.
Abstract:
- ‘We developed a therapeutic education program (TEP) aiming at helping them achieving this goal and report here its preliminary results.” What is “this goal”? Who were “them”
- “A working group of 15 health professionals developed a structured TEP for patients with CVI adapted to the regular French spa 3-week treatment course with the help of a group of patients. ”The patients with CVI and the help of a group of patients: were they the same group of patients with CVI? The sentence is unclear.
- What is CIVIQ2?
- Mean age 69 years old (SD 8 years) is more appropriate
- P=0.0024? and 0.018?
- What do you mean “opening the way for the development of similar programs in setting of spa resorts”? You meant the similar intervention is necessary to be promoted?
- However, it is not unclear how knowledge, skills and motivation improve health-related behavior. The word limit of the journal should be 300 words
Introduction
- This section aims to let the readers understand your topic and what the knowledge gap or why you conduct this study. However, the description is inadequate.
- There are very long sentences that affect the understand and made the readers lose the focus.
- More elaboration for the study purpose and knowledge gap is essential.
Materials and Method
- The author should check the format of the manuscript before submission. 2.1 patients should be started another new line.
- The first paragraph started with a very long sentence
- What is “They “on line 105 on page 3
- What are (28C) on line 107 and (34C) on line 114? You meant oC?
- How was the spa intervention designed including its formats, mode of the sessions, and duration
- Inadequate analysis
Results
- The aim of this paper was to report our preliminary experience with this program and the effects observed on patient health-related behavior and quality of life.
- The results were reported inadequately.
Discussion
- This part is to interpret the results, however, this part is too short and inadequate report
- I could not see the results in Table 2 with a very high level of satisfaction and motivation. Please more elaboration
All in all, the paper was inadequately reported. English proofreading may be needed because there are multiple grammatical mistakes.
Author Response
Thank you very much for your comprehensive and patient evaluation of our work, as well as your many valuable suggestions. We made many changes in the manuscript, and you will find below the answers to your shrewd questions.
Abstract:
- ‘We developed a therapeutic education program (TEP) aiming at helping them achieving this goal and report here its preliminary results.” What is “this goal”? Who were “them”
Since the word limit of the journal is 300 and not 200 as we thought, there is room for a bit more explanations, and we rephrased some sentences in the abstract according your suggestions. In this particular sentence, "them" meant the patients and "this goal" was the optimization of their health related behavior. The new wording s : "As this is usually challenging for the patients, we developed a therapeutic education program (TEP) aiming at helping them to achieve these optimizations.”
- “A working group of 15 health professionals developed a structured TEP for patients with CVI adapted to the regular French spa 3-week treatment course with the help of a group of patients. ”The patients with CVI and the help of a group of patients: were they the same group of patients with CVI? The sentence is unclear.
The program is dedicated to the many patients with CVI, but a small group of patients reviewed and amended the proposals of the working group of health professionals. We suppressed the mention of these helping patients in the abstract because it is not a major information and seems confusing. We also rephrased it: "This structured TEP was developed by a working group of 15 health professionals to be used during the regular French spa 3-week treatment course for CVI patients. ”
- What is CIVIQ2?
CIVIQ2 is a quality of life instrument dedicated to CVI and well validated and recognized in the vascular world. We expanded a little bit its description in the abstract and so more in the methods section.
- Mean age 69 years old (SD 8 years) is more appropriate
- P=0.0024? and 0.018?
We agree with these changes.
- What do you mean “opening the way for the development of similar programs in setting of spa resorts”? You meant the similar intervention is necessary to be promoted?
Yes, exactly. Indeed, the Spa resorts are very well suited for such educational programs, and we stated it clearly in the discussion section of the revised manuscript.
- However, it is not unclear how knowledge, skills and motivation improve health-related behavior.
As a matter of fact, many patients with CVI do not know very well what themselves can do to actively participate in their treatment, even if the improvement of health-related behaviors is important as shown in many epidemiological studies. But even when they know what they can do, succeeding in changing behaviors is not straightforward for many people, and require a high level of motivation. This is what the program allows. We mentionned these explanations in the discussion section and also in the introduction of the revised manuscript
- The word limit of the journal should be 300 words
This is great news, and we took better advantage of it.
Introduction
- This section aims to let the readers understand your topic and what the knowledge gap or why you conduct this study. However, the description is inadequate.
- There are very long sentences that affect the understand and made the readers lose the focus.
- More elaboration for the study purpose and knowledge gap is essential.
In the revised manuscript, we clarified the explanation of the rational and insist on the novelty of this approach for CVI patients.
Materials and Method
- The author should check the format of the manuscript before submission. 2.1 patients should be started another new line.
You are right: it is a "typo" mistake that we corrected
- The first paragraph started with a very long sentence
- What is “They “on line 105 on page 3
"They" was for "These sessions". We made it clear
- What are (28C) on line 107 and (34C) on line 114? You meant oC?
Yes, this is how it appears on our computer ; we wrote the full words in the revised manuscript
- How was the spa intervention designed including its formats, mode of the sessions, and duration
The spa intervention is the tailored by the physician according to the patients capabilities ; the typical regimen with 4 care sessions is described. We feel the rephrasing of the first sentence makes it clearer in the new version
- Inadequate analysis
We tried to make even clearer that it was a description, not a demonstration, but in any case, this short paragraph describes what had be done
Results
- The aim of this paper was to report our preliminary experience with this program and the effects observed on patient health-related behavior and quality of life.
- The results were reported inadequately.
The core results are in paragraph 3.4. but we think it is important to describe first the patients who were treated and the practical course of the treatment before the main results. We changed the title of the main results section to show it is the most important
Discussion
- This part is to interpret the results, however, this part is too short and inadequate report
We understand
- I could not see the results in Table 2 with a very high level of satisfaction and motivation. Please more elaboration
You are right that there is no direct question about satisfaction to the participants ; however, we consider that 75% full agreement and 25% "rather yes" agreement with "I am happy with my participation being happy with his participation to the program" is a very high level of satisfaction. In the same way, 94% thinking they will be more active in the management of their disease is indeed a clear sign of high motivation; don’t you think so? We improved a little bit the translation of the third sentence which was originally in French of course to make it closer to the real meaning, ("live well with my disease").
All in all, the paper was inadequately reported. English proofreading may be needed because there are multiple grammatical mistakes.
We feel that probably the length of some sentences made it look like as with approximative grammar. But we are ready to make any correction if you could give us more precise directions about where these mistakes take place.
Thank you again for taking your precious time helping us to improve this manuscript which is important to us, and, we hope, to the many patients that might benefit from this therapeutic approach.